# Potential Use of *Agave* Genus in Neuroinflammation Management

**DOI:** 10.3390/plants11172208

**Published:** 2022-08-25

**Authors:** Maribel Herrera-Ruiz, Enrique Jiménez-Ferrer, Manasés González-Cortazar, Alejandro Zamilpa, Alexandre Cardoso-Taketa, Martha Lucía Arenas-Ocampo, Antonio Ruperto Jiménez-Aparicio, Nayeli Monterrosas-Brisson

**Affiliations:** 1Centro de Investigación Biomédica del Sur, Instituto Mexicano del Seguro Social (IMSS), Xochitepec 62740, Mexico; 2Centro de Investigación en Biotecnología, Universidad Autónoma del Estado de Morelos (UAEM), Cuernavaca 62209, Mexico; 3Centro de Desarrollo de Productos Bióticos, Instituto Politécnico Nacional (IPN), Yautepec 62739, Mexico; 4Facultad de Ciencias Biológicas, Universidad Autónoma del Estado de Morelos (UAEM), Cuernavaca 62209, Mexico

**Keywords:** *Agave*, neuroinflammation, saponins, cantalasaponin

## Abstract

Agavaceae contains about 480 species, commonly used in the production of alcoholic beverages such as tequila and mezcal, making it a resource of economic and cultural importance. Uses of this plant rely mainly on the stem; other components such as the leaves are discarded, generating agro-industrial waste, despite being a source of bioactive and nutraceutical products. Reports show anti-inflammatory and anti-neuroinflammatory effects of these species, with flavonoids and saponins being mainly responsible. Neuroinflammation is a brain process that plays a key role in the pathogenesis of various neurodegenerative disorders and its effects contribute greatly to mortality and morbidity worldwide. This can be triggered by mechanisms such as glial reactions that lead to the release of inflammatory and oxidative molecules, causing damage to the CNS. Treatments do not cure chronic disease associated with inflammation; they only slow its progression, producing side effects that affect quality of life. Plant-based therapy is promising for treating these diseases. Pharmacological activities have been described for the Agavaceae family; however, their role in neuroinflammation has not been fully investigated, and represents an important target for study. This review synthesizes the existing literature on the biologically active compounds of *Agave* species that are related in some way to inflammation, which will allow us to propose a line of research with this genus on the forefront to orient experimental designs for treating neuroinflammation and associated diseases.

## 1. Introduction

The Agavaceae Endl. Family is distributed in the American continent, among the United States, Central America and the Antilles [1]. In Mexico, approximately 342 species are recognized, and are distributed among eight different genera: *Agave* L., *Beschorneria* Kunth, *Furcraea* Vent., *Hesperaloe* Engelm., *Manfreda* Salisb., *Polianthes* L., *Prochnyanthes* S. Watson and *Yucca* L. [2]. The *Agave* genus is mostly distributed in Mexico, where approximately 75% of these species are located [3].

In Mexico, the genus *Agave* is of great commercial and cultural importance; several species have been used for centuries as a source of fiber, food, medicine, fuel, shelter, ornament, textiles, compost and alcoholic beverages such as tequila, mezcal and bacanora [4]. More recently, the interest in these species has been growing as potential nutraceuticals, prebiotics, natural sweeteners and biofuels [5,6,7].

In traditional medicine, *Agave* species have been used to treat wounds, sores, trauma, fractures, rheumatoid arthritis, psoriasis, snake bites [8,9], syphilis, scurvy, cancer, limb paralysis, and postpartum abdominal inflammation, as well as being used as diuretics and laxatives [10]. They have also been employed to treat diseases of bacterial etiology such as gastrointestinal and wound infections, urologic disorders, and dysentery as well as cancer, diabetes and hypertension [11]. 

There are several reports indicating that Agaves have a variety of medicinal properties such as antioxidant, antibacterial [11], anticancer [12] and anti-inflammatory [13,14] activities; at least one study reports an anti-neuroinflammatory effect [15]. Since the process of neuroinflammation can be a start point of several chronic neurodegenerative diseases [16,17,18,19,20], a huge number of natural products have been investigated regarding their capacity to act as anti-inflammatory agents, especially in the brain. However, the role in neuroinflammation has not been fully investigated for these species. The objective of this review is to frame the pharmacological studies that refer to the capacity of extracts and secondary metabolites isolated from several species of the genus *Agave*, to act on some pathophysiological process associated with inflammation, so that the reader can visualize the potential of this genus and its unexplored applications in the field of neuroinflammation. In addition, the study of these plants could provide evidence for their employment in the treatment of this medical condition and related disorders. Therefore, studies that demonstrate various pharmacological effects are cited, highlighting antioxidant, immunomodulatory, anti-inflammatory and anti-neuroinflammatory properties. This review leads us to a description of the agaves, trying to address inflammation-related activities; this allows us to infer that this genus could become an object of study in the search of an anti-neuroinflammatory treatment, and for the development of a possible phytomedicine that could help to improve the quality of life of patients with chronic degenerative diseases with brain inflammation as a pathophysiological background. Additional value is assigned to the leaves of these species, because despite possessing a wide variety of active metabolites, they are considered as waste material in the mezcal and tequila industry, since the stem is the only material used.

## 2. Methodology

### 2.1. Search Planning

The methodology applied in this review consisted mainly in searching pharmacological information of species of the *Agave* genus, with reports about applications related to inflammation, in databases such as PubMed, Google scholar and Scopus. Showing the following results for the queries “anti-inflammatory *Agave* polysaccharide” (2); “anti-neuroinflammatory *Agave* polysaccharide” (0); “anti-inflammatory *Agave* saponins” (195); “anti-neuroinflammatory *Agave* saponins” (1); “anti-inflammatory *Agave* steroidal saponins” (141); “anti-neuroinflammatory *Agave* steroidal saponins” (1); “anti-inflammatory *Agave* terpenes” (102); “anti-neuroinflammatory *Agave* terpenes” (0); “anti-inflammatory *Agave* alkaloids” (200); “anti-neuroinflammatory *Agave* alkaloids” (0); “anti-inflammatory *Agave* flavonoids” (2); “anti-neuroinflammatory *Agave* flavonoids (0)”; “anti-inflammatory *Agave* fatty acids” (258); “anti-neuroinflammatory *Agave* fatty acids” (6); “anti-inflammatory *Agave* β-sitosterol” (1) and “anti-neuroinflammatory *Agave* β-sitosterol” (0); “neuroinflammation and Isorhamnetin” (7); “neuroinflammation and hesperidin” (31); “neuroinflammation and delphinidin” (0); “inflammation and delphinidin” (60); “neuroinflammation and quercetin” (115); “neuroinflammation and apigenin” (48); “neuroinflammation and catechin” (90).

### 2.2. Study Selection

Once the search was complete, it was apparent that studies focusing on evaluating the anti-neuroinflammatory activity of *Agave* compounds were scarce. However, there are studies that report anti-inflammatory activity in this genus and they could possibly act at the CNS level. Due to this, our bibliographic selection method consisted in choosing all those studies related with *Agave* genus pharmacology and phytochemistry, without considering the date of publication. 

### 2.3. Data Extraction

The selected papers were divided into three groups, those related to botanical and taxonomic information of the genus, phytochemical studies and finally, those related to pharmacological reports. 

The information gathered was synthesized in a table that included the following data: (a) scientific name, (b) common names, (c) medicinal uses, (d) distribution in Mexico and e) references. Biological activities reported for each *Agave* species mentioned in the text, as well as the compounds isolated, identified and/or evaluated, were synthesized for this work. Therefore, despite the number of studies that the query of different natural compounds accompanied by the word “*Agave* anti-inflammatory or anti-neuroinflammatory” showed, only those that actually addressed the natural compound type and pharmacology were analyzed. This is because most of the studies issued only appeared because of the similarity of some of the search terms. Hence, fatty acids, triterpenes and alkaloids will not be described in this review.

Finally, we selected only information useful to achieve the stated objective and decided to write the paper.

## 3. Pharmacological Background of *Agave* Species

This section includes background information related mainly to the pharmacology of the extracts of the *Agave* genus species, as well as some of the isolated compounds. In addition to the species mentioned here, the use of these plants in traditional medicine and their distribution in this country were also included (Table 1). This information demonstrates that the species of the *Agave* genus are perceived with important properties to reduce inflammation and that this attribute can be used as a point of research that analyzes in detail the anti-neuroinflammatory properties of these species that can be applied in the ailment of chronic degenerative diseases related to this process.

### 3.1. Antioxidant Activity

Using the in vitro methods of the free radical 2,2-diphenylpicrylhydrazyl (DPPH) and the 2,20–azinobis (3-ethylbenzothiazoline-6-sulfonic acid) (ABTS), the antioxidant capacity was measured (as µM of Trolox Equivalents (TE)/g dw)., for the methanolic extract of leaves from *A. rzedowskiana* (27.4127,41 ± 3.35 µM TE/g dw) > *A. ornithobroma* > *A. schidigera* > *A. tequilana* > *A. impressa* > *A. angustifolia* [11].

The *A. marmorata* Roezl chemical and pharmacological activity studies, showed that the saponin smilagenin-3-*O*-[β-D-glucopyranosyl (1→2)-β-D-galactopyranoside] **(1)** inhibited the production and release of nitric oxide induced with lipopolysaccharide (LPS) (EC_50_ = 5.6 mg/mL, E_max_ = 101%) [29]. 

For *A. salmiana*, it was determined that the culture conditions IN (in vitro plants obtained from the tissue culture laboratory at the multiplication step), EN (ex vitro plants obtained after acclimatization step) and WT (wild-type plants obtained from a natural population) exerted an effect on the total concentration of saponins and phenolic acids in methanolic leaf extracts. They observed an increase of 35 and 40% in the phenolic acids content in the IN and EN plants, respectively, compared with WT. The same pattern was observed in terms of total saponin concentration; IN and EN plants showed a 36- and 29-fold increase in saponin content compared to the WT plants. Antioxidant activity was determined using the oxygen radical absorbance capacity assay. IN had the highest antioxidant activity (369 µmol TE/g dw), followed by EN and WT, (184 and 146 µmol TE/g dw). The IN activity was 2.5-fold and 2.8-fold higher than EN and WT, respectively. However, a correlation between antioxidant activity and total phenolic acids or total saponins was not found [30]. 

The methanolic extract of *A. americana* leaves showed a direct correlation between total polyphenol and total flavonoid contents on antioxidant activity. According to their capacity to neutralize the free radical 2,2-diphenylpicrylhydrazyl (DPPH), the best one was that with a high total polyphenol content [31].

### 3.2. Anti-Inflammatory and Immunomodulatory Activities

*A. angustifolia* extracts, fractions and a purified compound 3-*O*-[(6’-*O*-palmitoyl)-β-D-glucopyranosyl sitosterol (**2**), isolated from *A. angustifolia,* have demonstrated an anti-inflammatory effect against the oedema induced in mouse ears by 12-*O*-tetradecanoylphorbol-13-acetate (TPA) at 0.8 mg/ear. Treatment with *A. angustifolia* led to a decrease in the concentrations of various pro-inflammatory cytokines such as Tumor Necrosis Factor-α (TNF-α), Interleukin-1β (IL-1β) and Interleukin-6 (IL-6) as well as an increase in anti-inflammatory cytokines Interleukin-4 (IL-4) and Interleukin-10 (IL-10) [14]. 

*A. tequilana* has been shown to have an immunomodulatory effect in a model of systemic lupus erythematosus that is experimentally induced with pristane (2,6,10,14-tetramethylpentadecane). *A. tequilana* extract decreased articular inflammation and proteinuria, reduced ssDNA/dsDNA antinuclear antibody titers, reduced IL-1β, IL-6, TNF-α, Interferon-γ (IFN-γ) levels and increased IL-10. The phytochemical analysis of *A. tequilana* extracts revealed the presence of β-sitosterol glycoside, phytol, octadecadienoic acid-2,3-dihydroxypropyl ester, stigmasta-3,5-dien-7-one, cycloartenone and cycloartenol [32,33,34]. Aqueous extract and acetonic fractions of this plant, administered in mice (50 mg/kg) with hypertension induced by AGII, showed the plant’s capacity to diminish that condition, and also provoked a significant decrease in IL-1β, IL-6 and TNFα kidney levels induced by AGII. *A. tequilana* methanolic fraction showed TNFα levels in the kidney similar to the baseline treatment [35].

The saponin (**1**) isolated from *A. marmorata*, evaluated on a culture of RawBlue cell line, was capable of inhibition of NF-κB expression (EC_50_ = 0.086 mg/mL, E_max_ = 90%), [29].

Dry extracts from decoctions of *A. intermixta* leaves (300 and 500 mg/kg p.o.) showed potent in vivo anti-inflammatory activity in the carrageenan-induced rat paw edema with a 50.13 ± 3.8% edema inhibition, meanwhile in an assay of TPA-induced ear edema, the topical administration of *A. intermixta* leaves decoctions (3 and 5 mg/ear) significantly inhibited oedema (50%) compared to the control group [36].

The aqueous extract of *A. americana* was shown to contain terpenoid compounds and steroidal saponins with anti-inflammatory properties, including agavasaponin E (**3**) and agavasaponin H (**4**), hecogenin (**5**) and tigogenin (**6**) [37,38,39,40,41,42,43]. Using the carrageenan-induced oedema model, the aqueous extract (39 µg/kg) from *A. americana* and a sapogenin-enriched fraction extract (hecogenin and tigogenin) showed anti-inflammatory effects, with 50% and 70–100% inhibition of oedema, respectively [37]. *A. americana* hydroalcoholic extract (100, 200 and 400 mg/Kg) caused a significant concentration-dependent reduction in paw oedema [43]. 

The pharmacologic effect of cantalasaponin-1, a saponin isolated from the species *A. americana, A. barbadensis* [44] and *Furcraea selloa* var. *marginata* [45], was evaluated on TPA-induced auricular oedema, showing a dose-dependent inhibitory effect of up to 90% at the highest dose of 1.5 mg/ear [13].

It should be noted that in the review carried out in the literature, there is only one report about anti-neuroinflammatory activity to the genus *Agave*, specific to *A. americana* and the compound cantalasaponin-1. In this study, *A. americana* (125 mg/Kg) and cantalasaponin-1 (5 and 10 mg/kg) reduced brain concentration of LPS-induced proinflammatory cytokines IL-6 and TNF-α. Cantalasaponin-1 increased the brain concentration of the anti-inflammatory cytokine LI-10 [15].

### 3.3. Anti-Cancer and Anti-Bacterial Activities

The anticancer activity of linear inulin-type fructan (ITF) prebiotics from *A. angustifolia* play a significant role in the prevention of colorectal cancer. This was evaluated by using a culture of the human colon cancer cell line Caco-2 (colorectal adenocarcinoma), in the Simulator of Human Intestinal Microbial Ecosystem model. The proximal, transverse and distal vessels were used to investigate fructan fermentation throughout the colon to assess the alterations of the microbial composition and fermentation metabolites (short chain fatty acids and ammonia). The influence on bioactivity of the fermentation supernatant was assessed by MTT (viability assay), Comet (DNA damage assay) and transepithelial electrical resistance (TER), respectively. *A. angustifolia* fructans significantly increased the population of bifidobacteria, short-chain fatty acid levels and transepithelial electrical resistance while decreasing ammonia levels. This indicates a protective effect on the function of the intestinal barrier, which is an important aspect in the pathology of colon cancer [46].

Five saponins isolated from *A. fourcroydes* methanolic leaves extract were evaluated to determine cytotoxic activity by fluorometric microculture cytotoxicity assay (FMCA). A new steroidal saponin, elucidated as chlorogenin 3-*O*-[α-L-rhamnopyranosyl-(1→4)-β-D-glucopyranosyl-(1→3)-{β-D-glucopyranosyl-(1→3)-β-D-glucopyranosyl-(1→2)}-β-D-glucopyranosyl-(1→4)-beta-D-galactopyranoside], along with four known saponins including furcreastatin, chlorogenin 3,6,-di-*O*-β-D-glucopyranoside and tigogenin3-*O*-[α-L-rhamnopyranosyl-(1→4)-β-D-glucopyranosyl-(1→3)-{β-D-glucopyranosyl-(1→3)-β-D-glucopyranosyl-(1→2)}-β-D-glucopyranosyl-(1→4)-β-D-galactopyranoside] showed strong cytotoxicity against HeLa cells with IC_50_ values of 13.1, 5.2, and 4.8 µg/mL, respectively [47].

A smilagenin di-glycoside elucidated as (25*R*)-5*β*-spirostan-3*β*-yl*O*-*β*-D-glucopyranosyl-(1→4)-*β*-D-galactopyranoside and isolated from *A. utahensis* Engelm showed cytotoxicity against HL-60 cells with an IC_50_ value of 4.9 µg/mL, inducing apoptosis through a marked caspase-3 activation [48]. Steroidal saponins isolated from *A. utahensis* showed moderate cytotoxic activity against HL-60 cells at 20 µg/mL in contrast to the control, etoposide [49]. The spirostanol saponin AU-1 isolated from *A. utahensis*, caused a transient increase in cyclin-dependent kinase inhibitor (CDKI) p21/Cip1, through the upregulation of the miRNAs miR-34 and miR-21. AU-1 stimulated p21/Cip1 expression without exerting cytotoxicity against different types of carcinoma cell lines (human renal adenocarcinoma-derived ACHN cells and human hepatocellular carcinoma HepG2 cells). In renal adenocarcinoma ACHN cells, AU-1 transiently elevated the expression level of p21/Cip1 protein without marked increases in p21/Cip1 mRNA levels. Rapid and transient increases in miR-34 and miR-21, known to regulate p21/Cip1, were observed in AU-1-treated cells [50].

Concentrated *Agave* Sap (CAS) obtained from 18 different Mexican states, showed an antiproliferative effect on the culture of cancer cells since it significantly reduced cell viability to 80 % when tested at a concentration of 75 μg/mL on HT-29 (HTB-38), a cellular line with epithelial morphology isolated from colorectal adenocarcinoma. *Agave* sap also had apoptotic activity on HT-29 (IC503.8 ± 1.3 mg/mL) and NIH-3T3, which are embryonic mouse fibroblast, (IC508.4 ± 1.0 mg/mL) cell lines. The activities found were attributed to the main saponins detected in CAS, including kammogenin and manogenin [12].

The *n*-butanol extract from inflorescences of *A. schotti* Engelm was active against Walker carcinoma 256 (intramuscular) tumor system (5WM). Activity was detected in Sprague rats at a level of 7% T/C (Test Control; when the degree of screening activity is significantly greater than the minimum values: ≥125 for survival systems or ≤42 for tumor-weight inhibition) at 75 mg/kg and 28% T/C at 37.5 mg/kg, induced by the saponin-rich fraction separated from the *n*-butanol extract. Six saponins were isolated, the fourth being the most active against the 5WM tumor system (17% T/C at 65 mg/kg and 22% T/C at 33 mg/kg). Antitumor activity of the system was defined as a percent T/C value of less than 60 in a satisfactory dose response test [51].

Methanolic extract of *A. tequilana* leaves had antibacterial activity against *Pseudomona aeruginosa* ATCC 27853 and *Escherichia coli* ATCC 25922 strains (Minimal Inhibitory Concentration of 5 mg/mL) and the evaluation was carried out using the broth microdilution method. These results were associated with the presence of metabolites like tannins, alkaloids, flavonoids, and saponins [11]. *A. tequilana* syrup at 50% (*w*/*v*) inhibited *Bacillus subtillis* 168 and *E. coli* DH5 growth. The bacteriostatic activity was attributed to the high sugar content and high viscosity [52].

The *A. americana* methanolic leaf extract induced a potent cytotoxic effect against MCF-7 (breast carcinoma) and Vero (African green monkey kidney) cell line. IC_50_ values were found to be 546 and 1854 μg/mL respectively by the SRB (determination of total cell protein content by sulphorhodamine B) assay [53].

### 3.4. Other Activities of Agave Genus

It has been established that obesity is associated with chronic inflammation, and it has been demonstrated in animal models that *A. tequilana* fructans intake induced a decrease in body mass index (35.3 kg/m^2^ to 33.0 kg/m^2^), total body fat percentage (38 to 20%), and also in triglyceride levels (167.4 to 107.9 mg/dL *p* < 0.05), in obese individuals [54].

In addition, the aqueous extract of *A. salmiana* leaf (100 mg/Kg) has anthelmintic properties, strongly reducing *H. gallinarum* worm egg counts and worms [55]. Hecogenin (**5**) (90 mg/Kg), a steroid saponin isolated from *A. salmiana* leaves, showed anti-ulcer properties in ethanol- and indomethacin-induced gastric ulceration. The gastroprotection mechanism of hecogenin was K-ATP channel dependent [56]. In another study, the depressive behavior induced by reserpine (2 mg/kg, o.p.) was reversed by hecogenin (5 and 10 mg/kg), since it decreases the immobility time compared to the control group (Tween 80/4%) similarly to imipramine (10 mg/Kg) [57].

Magueyosides A, B, D and E isolated from the hydroalcoholic extract of *A. offoyana* flowers, showed phytotoxic activity at concentrations above 100 µM. They significantly inhibited *Lactuca sativa* L. root growth (IC_50_ 88.4µM, 104.3 µM, 131.2 µM and, 101.6 µM, respectively) compared to the commercial herbicide Logran^®^ (523.7 µM). At the same time, saponin concentrations below 33 µM enhanced root growth [1].

Two saponin-enriched fractions (SFs) obtained through the best traditional extraction method (ethanol:water 8:2) and the best new method (*n*-butanol:water 1:1) from *A. tequilana, A. salmiana, A. angustifolia, A. furcroydes, A. hookeri, A. inaequidens, A. marmorata and A. atrovirens* leaves exhibited some selective phytotoxic activity against weeds from the Standard Target Species (STS) *Solanum lycopersicum* (tomato), *Lactuca sativa* (lettuce) and *Lepidium sativum* (cress), as well as on two harmful weed species for agricultural crops *Lolium perenne* (perenne ryegrass) and *Echinochloa crus-galli* (barnyardgrass). The authors mentioned that the saponins from the leaves of these species could be a potential source of ecoherbicides [58].

The insecticidal and repellent activity of *A. americana* methanolic leaf extract was also assessed either by topical application, treated filter-paper methods or repellent bioassays against *Sitophilus oryzae* adults. The activity was attributed to the main identified compounds (flavonoids and saponins) permeating the insect’s cuticle [59].

The steroidal saponins Agamenoside G, degalactotigonin, cantalasaponin 1, (25*R*)-5α-spirost-3β-hydroxy-12-oxo-3-*O*-β-D-glucopyranosyl-(1-2)-[β-D-xylopyranosyl-(1-3)]-β-D-glucopyranosyl-(1-4)-β-D-galactopyranoside, deltonin, and dioscin isolated from *A. americana*, showed antifungal activity against *Candida albicans* ATCC 90028, *Candida glabrata* ATCC 90030, *Candida krusei* ATCC 6258, *Cryptococcus neoformans* ATCC 90113, and *Aspergillus fumigatus* ATCC 90906 [60].

## 4. Natural Products from *Agave* Species

In the next section, we present the names of some natural compounds that are part of the secondary metabolism of *Agave* species (Table 2). According to the purposes of this review, we present those that we consider the most relevant in terms of the anti-inflammatory and/or anti-neuroinflammatory activity they exert.

A number has been assigned to each compound. The structures correspond to compounds from different chemical families.

### 4.1. Phytosteroles

Phytosterols are bioactive compounds that are naturally present in plant cell membranes with chemical structures similar to the mammalian cell-derived cholesterol. Among various phytosterols, β-sitosterol (**7**) is the main compound, found in abundant quantities in plants. It has been evidenced in many in vitro and in vivo studies that SIT possesses various biological functions such as anxiolytic, sedative, analgesic, immunomodulator, antimicrobial, anticancer, anti–inflammatory, lipid lowering effect, hepatoprotective, aid in respiratory diseases, wound healing effect, antioxidant and anti-diabetic activities [61].

#### 4.1.1. β-Sitosterol and Neuroinflammation

BV2 murine microglial cells treated with different concentrations of SIT prior to LPS stimulation, showed that (**1**) reduced the LPS-induced expression of inflammatory mediator like IL-6, inducible nitric oxide (iNOS), TNF-α and cyclooxygenase-2 (COX-2). SIT treatment also inhibited the LPS-induced activation of 38p, ERK and Nuclear factor kappaB (NF-κB) signaling pathways [62].

#### 4.1.2. *Agave* β-Sitosterol

The acetonic extract of *Agave*
*angustifolia* stem showed immunomodulatory activity and it was demonstrated that the compounds responsible for this effect were β-sitosterol β-D-glucoside (BSSG) and its free phytosterol, known as β-sitosterol [14].

### 4.2. Flavonoids

Flavonoids are phenolic compounds comprising fifteen carbon atom structures (C6-C3-C6). They consist of two aromatic carbon group rings: benzopyran and benzene. According to their structure, they are classified as flavones, flavonols, flavanones, flavanonols, flavanols, anthocyanidins, isoflavones, neoflavonoids and chalcones [63].

#### 4.2.1. Flavonoids and Neuroinflammation

Recent studies demonstrate the potential role of these compounds in the treatment of neurodegenerative disorders.

#### 4.2.2. *Agave* Flavonoids

HPLC-UV-MS analyses confirms the concentration of isorhamnetin (**8**, 1251.96 μg), flavanone (291.51 μg), hesperidin (**9**, 34.23 μg), delphinidin (**10**, 24.23 μg), quercetin (**11**, 15.57 μg), kaempferol (13.71 μg), cyanidin (12.32 μg), apigenin (9.70 μg) and catechin (7.91 μg) per gram of *Agave lechuguilla* dry residue [64]. Hamissa et al. (2012) reported that the flavonoid content in *A. americana* leaves ranged from 0.96 to 4.90 mg of quercetin equivalents g^−1^ d.w. Gallic acid was also reported in *A. tequilana, A. ornithobroma, A. impressa, A. rzedowskiana, A. schidigera* and *A. angustifolia* [31]. Tannic acid was reported from *A. americana* leaf extract [65]. In *A. durangensis* extract, several flavonoids were identified, including kaempferol glycoside, quercetin glycoside, kaempferol-3,7-*O*-diglucoside, kaempferol-3-*O*-[6-acetylglucoside]-7-*O*-glucoside, kaempferol-3-*O*-[rhamnosyl(1-6) glucoside], quercetin-3-*O*-arabinoside and kaempferol-3-*O*-rhamnoside [66].

Several flavonoids have been shown the ability to exert pharmacological effects on the CNS. Many of these have been isolated in different *Agave* species, but there are no data in the bibliography that indicate this effect in flavonoids of this genus. We will now review some studies directly related to neuroinflammation, carried out with different flavonoids.

***Isorhamnetin ***(**8**). It was administered for 14 weeks in obese mice on a hypercaloric diet, and it was observed that it prevents cognitive deterioration and inhibits the overactivation of microglia and the high concentration of inflammatory cytokines in the serum and brain of said mice, in addition to decreasing the expression of inflammation regulators such as p-NFkB, p-JNK (c-Jun N-terminal kinase) and p-p38 (mitogen-activated protein kinase) [67]. Compound (**2**) exerts a concentration-dependent effect (50, 100 and 200 µM) on the effects of LPS (100 ng/mL) in BV2 microglia culture, inhibiting NO, prostaglandin E2 (PGE2), TNF-α, IL-1β concentration and inhibition of iNOS, COX 2, TNF α and IL-1β expression by ISO in BV2 microglial cells [68].

The neuroprotective effect of (**8**), was also evidenced in N9 cells culture, since it inhibited NO synthesis at an IC_50_ 17.87μM [69].

For ***Hesperidin ***(**9**), a flavanone derived from the secondary metabolism of fruits, mainly those belonging to the citrus class, a wide range of information is available about its neuroprotective effect, which has been analyzed in detail in several review papers [70,71]. Therefore, only the most recent data directly related to neuroinflammation will be indicated here.

LPS administration for several weeks in adult mice induces glial activation, synaptic dysfunction, cognitive impairment and neuroinflammation, among other factors. In addition, it induces the alteration of some mechanisms such as increased expression of markers such as Toll-like receptor-4 (TLR4), GFAP and Iba-1 in hippocampal and cortical regions. In this study, animals treated with (**9**), showed significantly reduced levels of these factors. Furthermore, the expression of phosphorylated nuclear factor-κB (p-NF-κB), TNF-α and IL-1β in both cortical and hippocampal regions, as well as in BV2 cells, decreased with the administration of (**9**). It also decreased reactive oxygen species generation and lipid peroxidation, as well as preventing apoptosis, improves synaptic integrity, cognition and memory processes [72].

Oral administration of (**9**), at 50, 100 and 200 mg/kg for 25 days in mice with experimental autoimmune encephalomyelitis (EAE), causes a significant decrease in the concentration of proinflammatory cytokines, such as IL-17, TNF-α and IL -6; as well as an increase in IL-10 and Transforming Growth Factor-β (TGF-β) in splenocytes and lymph nodes, while in the brain it was observed that (**3**) induces a decrease in the expression of the same inflammatory cytokines and an increase in anti-inflammatory ones. In this study, the authors point out that the neuroprotective effect of this flavonoid is through the induction of the polarization of CD4^+^ T cells towards regulatory T cells [73]. Oral administration of (**9**) at 100 and 200 mg/kg also has neuroprotective effect against sodium fluoride (NaF)-induced neurotoxicity in rats by reducing lipid peroxidation and increasing superoxide dismutase (SOD), catalase (CAT) and glutathione peroxidase (GPx) activities. In addition, this flavonoid reduced the levels of NF-κB, IL-1β, TNF-α, Beclin-1 (a mammalian protein that plays a central role in autophagy, a programmed cell survival process, which increases during periods of cellular stress and is extinguished during the cell cycle), LC3A and LC3B (structural proteins of autophagosomal membranes), in brain tissue [74].

***Delphinidin*** (**10**). This anthocyanin has been investigated in multiple pharmacological studies; however, when the term “neuroinflammation” is included in the search together with this compound, there are no results. Nevertheless, there are data indicating, for example, that it is able to reduce inflammation induced by spinal cord injury in rats, because (**10**) is able to reduce the levels of TNF-α, IL-6, COX-2, PGE2 and caspase-3 and levels of NF-κB, AP-1 and p38-MAPK expression [75].

***Quercetin*** (**11**). This flavonoid is one of the most studied; when its name is consulted in PUBMED, a total of 23,850 citations are generated. Delimiting it with “quercetin + neuroinflammation”, the result was 115 publications. As the aim of this paper is not to review this molecule and its anti-neuroinflammatory effect, only the three most recent ones and those in which the research model employed LPS will be mentioned.

LPS i.p. administration at 0.25mg/kg per day in C57BL/6N mice induces a wide spectrum of pathophysiological conditions, leading to neuroinflammation, oxidative stress and cognitive impairment, among others. In this scheme, intraperitoneal administration of (**11**) at 30 mg/kg per day for two weeks reduced gliosis and protected against neuroinflammation by causing a decrease in TNFα, COX-2 and NOS-2 expression, all in the cortex and hippocampus of adult mice compared to the control group of mice. Immunofluorescence showed an attenuation of the number of IL-1β-positive cells compared to control mice in the same regions of the brain [76]. LPS was injected into the lateral ventricle of rats at 2 μL/min during 5 min, eliciting neuroinflammation and anxiety-like behavior evaluated in the elevated maze. Under this experimental scheme, administration of (**11**) at 10, 50 and 100 mg/kg for 21 days produces an anxiolytic effect as well as reducing the levels of proinflammatory markers associated with NF-κB activation such as IL-6, COX-2, with a marked increase in brain-derived neurotrophic factor (BDNF) and mRNA levels and a decrease in the level of iNOS mRNA [77].

A recently published research paper shows that (**11**) protects against the deleterious effect of LPS by decreasing NO production and the expression of IL-6, TNF-α and IL-1β in macrophage and microglia cultures, preventing polarization towards the M1 genotype. This molecule also reduced reactive oxygen species (ROS) production and the ability of LPS to activate the phagocytic capacity of microglia. Likewise, (**11**) acts in a balanced way in cell cultures, since it modulates the activation towards the M2 genotype because of the increased expression of its markers such as IL-10, HO-1, GCLC (catalytic subunit of glutamate-cysteine ligase), GCLM (glutamate-cysteine ligase modifying subunit), and NQO1 (NAD(P)H quinone oxidoreductase-1) by directly activating the AMP-activated protein kinase (AMPK) and Akt signaling pathways [78].

***Apigenin*** (**12**). This flavonoid is abundant in fruits and vegetables, with extensive studies that indicate potential pharmacological effects upon the inflammatory process, as well as extensive reports on anti-neuroinflammatory properties; for this reason, only some data relevant to the present review are mentioned here.

In a co-culture of neurons and glial cells stimulated with LPS (1 µg/mL) for 26 days from neonatal and embryonic rat cortex (Wistar strain) and treated with (**12**), it was demonstrated that this compound reduced the decline of neurons and astrocytes, as well as the reduction in the microglial activation and the CD68 marker expression of the M1 genotype. LPS or IL-1β (10 ng/mL) stimulated-cells increased IL-6, IL-1β and CCL5 mRNA expression and decreased IL-10 mRNA expression, actions that were counteracted by administering this flavonoid to the medium. Together, these activities were associated with the neuroprotective effect of this secondary metabolite [79].

After 22 days of a single injection of LPS (5 μg/5 μL) into the substantia nigra of rats, body weight and behavioral parameters were measured. Subsequently, the levels of inflammatory and oxidative markers, such as TNF-α, IL-1β, IL-6 and level of nitrite, MDA, and SOD, were evaluated in the striatum of these rats. NF-κB and Nrf-2 were measured by immunohistochemical analysis. The results indicated that these parameters were positively modulated by (**12**) oral administration at doses of 2.5 and 50 mg/kg [80].

***Catechin*** (**13**). From the flavan-3-ol family, it is widely recognized due its high concentration in tea leaves as well as its antioxidant and anti-inflammatory properties, among others that have been widely studied. In this review, we will mention a little information about the (**13**) effect on neuroinflammation, just as a background to highlight that polyphenolic compounds represent natural active compounds, with neuroprotective potential.

Stimulation of microglial cells with toxic agents such as LPS and Aβ (amyloid-β) peptide triggers their activation to produce inflammatory mediators, contributing to neurodegeneration and cell death. BV-2 cell culture activated with amyloid (5 µM of Aβ (1-42) for 12 h, leads to the production and overexpression of cytokines such as TNF-α, IL-1β, and IL-6 via regulation of NF-κB signaling pathway. iNOS expression levels were also increased. These damage variables were inhibited with (**13**) application (10 µM) in the culture medium [81].

### 4.3. Terpenes

Terpenes, terpenoids or isoprenoids, are grouped into natural product categories based on their structure and pathways for their biosynthesis. All terpenes are derived from the 5-carbon precursor compounds dimethylallyl diphosphate (DMAPP) and its isomer isopentenyl diphosphate (IPP), and exist as single unit hemiterpene (5C) to mono- (C10), sesqui- (C15), di- (C20), sester- (C25), tri- (C30), tetra- (C40) to polyterpenes of > C40–C5 × 10^3–4^ units. They are components of both the primary and secondary metabolism in cells. To date, tens of thousands of terpenes compounds have been identified, the majority encountered in organisms from the plant kingdom [82]. Many plant terpenes have found fortuitous uses in medicine, and the terpenes family of natural products has been a valuable source of medical discoveries [83].

#### 4.3.1. Terpenes and Neuroinflammation

Fourteen terpenes isolated from Lychee seeds (*Litchi chinensis* Sonn.) showed inhibitory effects on LPS-induced NO production in BV-2 cells, as they reduced COX-2, iNOS and IL-6 expression [84].

#### 4.3.2. *Agave* Terpenes

Several terpenes have been identified in extracts of *Agave* species. For example, α-linalool, α-terpinene, *p*-cymene, limonene, β-*trans*-ocimene, linalool, α-terpineol, geraniol and trans-nerolidol have been identified in *A. salmiana* and *A. angustifolia* extracts. In addition to the aforementioned terpenes, other terpenes have also been reported in *A. tequilana* Weber such as β-*cis*-ocimene, sabinene, 2,4,6-octatriene, 4-terpineol, α-terpineol, nerol, bornyl formate, α-cubebene, copaene, anastreptene, bergamotene, β-farnesene, 1,2,3,4-tetrahydronaphthalene, germacrene, α-curcumene, α-muurolene, α-bisabolene, cadinene, α-spirovetivene, cedrol, *trans*-nerolidol, cadalene, cadinol, patchouli alcohol and α-bisabolol [85]. There are many studies that have demonstrated various pharmacological activities such as the anti-inflammatory properties. However, when we ran the query for each one of these terpenic compounds found in *Agave* species, plus the word “anti-neuroinflammatory”, no results were found.

The variety of biological activities reviewed in the previous paragraphs emphasizes the potential use of terpenes and flavonoids in the treatment of disorders associated with neuroinflammation. It is important to realize that the lack of reports of anti-neuroinflammatory activity of these compounds in species of *Agave* genus does not detract from the novelty of the present study; on the contrary, the pharmacological background of these compounds in other species allows us to propose new research strategies focused on evaluating the biological anti-neuroinflammatory activity of the by-products of this genus species and their possible employment in the associated treatment of some disorders.

### 4.4. Saponins

Saponins are a heterogeneous group of glycosides that are widely distributed in plants of agricultural importance, particularly legumes [86]. They comprise a sugar moiety (glucose, arabinose, galactose, glucuronic acid, xylose, rhamnose or methyl pentose) linked to a non-polar aglycone (sapogenin) and are classified as steroids (C27) and triterpenoids (C30) [87].

#### 4.4.1. Saponins from *Agave* Species

They were first reported in the *Agave* genus in 1932 [88]. Since then, saponin and sapogenin constituents have been identified in approximately 50 species of this genus [89,90,91] and they are one of the most abundant compounds in *Agave* plants [3]. Steroidal sapogenins such as spirostanol-type and cholestane-type (only Agavegenin D) have been isolated from *Agave* leaves, flowers, callus cultures and rhizomes. Sapogenins with furostanol and furospirostanol skeleta have not been discovered in this genus yet [92].

There are many studies in which several sapogenins have been isolated in various species of *Agave*; among these are Hecogenin, Sisalagenin, Neohecogenin, Gloriogenin, Manogenin, 9-Dehydrohecogenin, 9-Dehydromanogenin, Botogenin, Gentrogenin, Kammogenin, Hainangenin, Hongguanggenin, Chlorogenin, Agavegenin A, Tigogenin, Neotigogenin, Gitogenin, Neotigogenone, Smilagenin, Sarsasapogenin, Ruizgenin, Diosgenin, Yamogenin, Yuccagenin, Agavegenin B, Agavegenin C, Agavegenin D. Furthermore, there are also reports of steroidal saponis such as Agavoside A, Chlorogenin 3-*O*-β-D-glucopyranoside, Chlorogenin 6-*O*-β-D-glucopyranoside, Agamenoside C, tigogenin 3-*O*-β-D-glucopyranoside, Agamenoside I among others. A great variety of Spirostanol diglycosides have also been reported, among which are Agavoside B and Cantalasaponin-1 and other spirostanol triglycosides, tetraglycosides, hexaglycosides and hexaglycosides, all of them with diverse biological activities [92].

Steroidal saponins are effective therapeutic options to combat inflammatory diseases because they are able to act directly on pro-inflammatory cytokines such as TNF-α and IL-6, inhibit the action of macrophages and act on the arachidonic acid (AA) pathway reducing COX-2 and PGE2 activity. They are receiving much attention from chemists and biologists for new drug discovery because they are strongly associated with anticarcinogenic, hepatoprotective, immunomodulatory properties, but especially anti-inflammatory and anti-neuroinflammatory activities in animal and in vitro models, since they are found in abundance in plants, which means that access to this type of drug would be much easier for the general population [93,94].

#### 4.4.2. Saponins and Neuroinflammation

It is well documented that saponins are neuroprotective compounds [95]. Gingsenosides Rg1 (**14**) and Re (**15**) exert neuroprotective effects in scopolamine-induced cognitive impairment, increasing choline acetyltransferase activity in rats [96], mice [97] and in vitro [98]. Compound (**14**) reduced ROS and cytotoxicity by downregulating the pro-apoptotic protein Bad, which counteracts oxidative stress, resulting in neuroprotection of LPS-treated cells [99]. Compound (**14**) also alleviates oxidative stress after ischemia/reperfusion (I/R)-induced neuronal injury by inhibiting miR-144 activity and blocking the Nrf2/ARE pathway at the post-translational level [100], promoting cerebral angiogenesis by increasing VEGF expression through the PI3K/Akt/mTOR signaling pathway after ischemic stroke. [101]. In addition, BV2 microglial cells pretreated with (**15**) exerted neuroprotective effects against LPS-induced neuroinflammation [102]. Compound (**15**) exhibited a potent neuroprotective effect against neuroinflammation in a murine model by reducing motor neuron death and decreasing pro-inflammatory cytokine TNF-α expression [103].

Pseudoginsenoside PF11 inhibited LPS-induced neuroinflammation in N9 microglia through the inhibition of TLR4 activation, NF-κB, iNOS, COX-2 expression and PGE2, IL-1β, IL-6 and TNF-α release [104].

Aster saponin B obtained from *Aster tataricus* L. root methanolic extract showed more than 50% inhibition of LPS-induced NO release in RAW264.7 cells and suppressed LPS-enhanced iNOS and COX-2 enzyme expression, resulting in the inhibition of pro-inflammatory mediators such as NO and PGE2. It also suppressed the phosphorylation and degradation of IκB in the cytoplasm. Therefore, the anti-inflammatory activity was exerted by inhibiting NF-κB activity in LPS-treated RAW264.7 cells [105].

Based on the review of the literature, many saponins have been isolated from the species of the *Agave* genus, however there is only one study that directly assayed a saponin extracted from *Agave* against neuroinflammation. In that study, the steroidal saponin cantalasaponin-1, isolated from the leaves of *Agave americana*, decreased the LPS-induced proinflammatory cytokines TNF-α and IL-6 and increased the anti-inflammatory cytokine IL-10 [15]. Considering that saponins are natural compounds with significant anti-neuroinflammatory activity [106], it is important to carry out more chemical and pharmacological studies in order to propose *Agave*s as medicinal plants that may be useful in the treatment of neuroinflammatory disorders.

In recent years, scientists have evaluated and discovered many medicinal plants and natural products. Many of them have been shown to be able to improve dementia treatment with fewer side effects than conventional drugs and are considered potential anti-neuroinflammatory drugs.the . Since the process of neuroinflammation may be the starting point of several neurodegenerative pathologies, a large number of natural products have been investigated for their ability to act as anti-inflammatories, especially in brain. In order to provide a complete perspective, in the following chapter, we will review some general aspects of some processes involved in neuroinflammation.

## 5. Neuroinflammation

Inflammation of the central nervous system (CNS), known as neuroinflammation, is a brain process that involves the blood–brain barrier, glia, and neurons and plays a key role in the pathogenesis of various neurodegenerative disorders such as Parkinson’s and Alzheimer’s diseases (AD) [106,107]. Collectively, these neuroinflammatory disorders and their sequelae contribute extensively to mortality and morbidity in the world with a significant healthcare cost [108]. Neuroinflammation can be triggered by various biological mechanisms including infection, trauma, ischemia, toxins, oxidative stress and glial reactions [109,110] and could be initiated in the periphery or within the brain [111].

The process is marked by the production of pro-inflammatory cytokines, including IL-1β, IL-6, IL-18 TNF-α, chemokines such as C-C motif chemokine ligand 1 (CCL1), CCL5 and C-X-C motif chemokine ligand 1 (CXCL1), small-molecule messengers, including prostaglandins, nitric oxide (NO), reactive oxygen species by innate immune cells in the CNS as well as increased levels of these molecules modulating the blood–brain barrier (BBB) [107].

### Neuroinflammatory Molecules

Cytokines are large proteins (15–25 kDa) that are mainly released from immune cells such as monocytes, macrophages and lymphocytes, in addition to microglia and astrocytes. They are activated during situations in which inflammation, infection and/or immunological alterations occur [112]. They are classified into two groups—pro-inflammatory and anti-inflammatory cytokines—which promote and inhibit inflammatory responses, respectively [113].

Among the pro-inflammatory cytokines, IL-1β, IL-6 and TNF-α are the most studied. IL-4 and IL-10 are well known anti-inflammatory cytokines. Under normal physiological conditions, pro-inflammatory cytokines are present at low levels [114], but in pathological conditions like infection or trauma, they can increase by up to 100-fold. Neuroinflammatory mediators such as cytokines and prostaglandins are important in the development of neurodegenerative diseases [115].

Initial release of cytokines can trigger the subsequent production of signaling molecules, as evidenced by IL-6, which activates T cells and stimulates the production of other inflammatory markers, such as C-reactive protein (CRP) and fibrinogen. Upon binding to the extracellular TNF-1 receptor, TNF-α sets in motion a series of signaling cascades that affect the transcription of genes related to apoptosis and neurodegeneration. IL-1β, when bound to the IL-1 receptor complex, is also a critical initiator of a number of signal transduction cascades, specifically the mitogen-activated protein kinase (MAPK) pathways, which result in many proinflammatory responses and the production of IL-8 and IL-6 [116].

Chemokines are small chemotactic cytokines that also play a role in neuroinflammation. Although they have very low physiological concentrations in the CNS, certain chemokines, such as monocyte chemoattractant protein-1, are strongly up-regulated in chronic neuroinflammation. These molecules are involved in the regulation and chemotaxis of some CNS cells such as astrocytes and microglia in response to an inflammatory stimulus and may also alter neuronal function and adversely affect neurogenesis [117].

The cyclooxygenase enzyme (COX) converts arachidonic acid into eicosanoid groups such as prostaglandins and thromboxane, and has various inflammatory functions. The pathways of its two common isoforms COX-1 and COX-2 are increasingly associated with neuroinflammation and neurodegeneration and COX inhibitors, such as nonsteroidal anti-inflammatory drugs (NSAIDs), have several therapeutic potentials [118].

## 6. CNS Cells

The CNS is populated by two broad categories of cells, neurons and neuroglial cells. The neuroglial cells include microglia, astrocytes, oligodendrocytes, and ependymal cells. The innate immune cells involved in neuroinflammation are primarily microglia and astrocytes but capillary endothelial cells and infiltrating blood cells also contribute to neuroinflammation, especially when the blood–brain barrier sustains biochemical or mechanical damage [119].

### 6.1. Blood–Brain Barrier (BBB)

The BBB is part of the neurovascular unit, which is essential for brain health and function. It is a dynamic interface between the peripheral circulation and the central nervous system, constituting a semi-permeable and highly selective structural and chemical barrier. Endothelial cells are the main structure of this barrier. Unlike other vascular endothelial cells lining peripheral blood vessels, cerebral microvascular endothelial cells exhibit distinctive morphological, structural, and functional characteristics that differentiate them from other vascular endothelia. These include the following: (1) the expression of tight junctions (TJs) that seal paracellular pathways between adjacent endothelial cells, thereby preventing the unregulated passage of polar (water-soluble) molecules between the blood and brain; (2) the absence of fenestrations; (3) the lack of pinocyticsynapto- activity and expression of active transport mechanisms to regulate the passage of essential molecules (including nutrients and essential amino acids), while blocking the passage of potentially unwanted substances (both endogenous and xenobiotic) [120].

Under optimal physiological conditions, the BBB ensures a highly stable internal brain environment and prevents foreign objects (such as microorganisms, toxins, etc.) from invading the brain tissue. However, under certain conditions such as brain edema, brain tumor, ionizing radiation injury, inflammation and other pathological conditions, BBB permeability increases, triggering acute or chronic neuroinflammatory processes that lead to neurodegenerative disorders [121].

### 6.2. Microglia

Microglia cells are the resident immune (macrophages) cells of the brain, and play a crucial role in the neuroinflammation process. Unlike neurons and other neuroglial cells which are derived from the neuroectoderm, microglia originate as primitive macrophages from the embryonic yolk sac and reach the CNS via the circulatory system [122]. Microglial cells are responsible for maintaining the tissue homeostasis of neuronal cells, which are sensitive to cell environment changes. In a non-stimulated state, the resting microglia feature a highly ramified appearance, which is in sharp contrast to the amoeboid reactive cells. They are characterized by rapid proliferation, the production and secretion of a wide spectrum of cytokines, chemokines and other immune mediators in response to insult [123].

In response to cytokines and other signaling molecules from acute inflammation, microglia transform from a ramified, inactivated state to an activated phagocytic one, releasing pro-inflammatory mediators in the process. They are activated in two ways: firstly, as a response to neural death caused by inflammatory neural injury, and secondly in response to viral or bacterial toxins. Microglial activation under homeostatic physiological conditions aims to constantly monitor the environment through long cytoplasmic extensions to activate the immune system and tissue repair in the CNS, as intrinsic and extrinsic factors can lead to disastrous dysregulation of microglial cells, resulting in severe CNS pathologies [124].

In terms of chronic neuroinflammation, microglia can remain activated for long periods releasing cytokines and neurotoxic molecules that contribute to long-term neurodegeneration by promoting their migration during the initial phase of neurodegeneration, attracted by factors secreted by cells that have already been injured. They are recruited to the damaged or infected area in order to participate in degenerative or regenerative processes, to engulf microorganisms and cell debris, and promote production of neuroinflammatory and neurotoxic molecules such as IL-1β, IL-6, TNF-α, prostaglandins, and nitric oxide (NO) [124]. Moreover, they activate FN-ĸB transcription factor to induce the expression of cytokine and chemokine genes, leading to neural cell death mediated by inflammatory processes [125].

### 6.3. Astrocytes

Astrocytes are the other glial cells within the CNS, which provide physiological functions related to CNS homeostasis such as regulation of blood flow, energy metabolism, immune defense, neurotransmission and neurogenesis. They also are highly heterogeneous in morphological appearance; they express a multitude of receptors, channels, and membrane transporters [126]. When any CNS damage occurs, such as neurodegeneration, trauma or infection, these cells are activated by a process called reactive astrogliosis, which express membrane receptors to recognize pathogens, depending on the intensity of the inflammatory process and damage [127].

An activated astrocyte secretes and/or responds to several cytokines, altering the neural cell state, microglia neighboring cells, and other astrocytes. Activation of these cells results in the synthesis and segregation of cytokines such as IL-1β, IL-6, TNF-α and transforming growth factor-β1 (TGF-β1). They promote or inhibit proinflammatory as well as anti-inflammatory gene expression [128].

With the term “neuroinflammation”, we refer to the inflammatory response in the CNS, as a result of neuronal injury and in related disorders that share a common neuroinflammatory pathology. BBB endothelium, microglia and astrocytes are key cellular drivers and regulators of neuroinflammation. Under physiological conditions, they are important for neurotransmission and synaptic homeostasis. However, an overexpressed inflammatory response mediated by these cells results in further cytokines production and a wide variety of inflammatory molecules that trigger synapto-toxicity and neurodegeneration in a self-reinforcing manner. Suppression of neuroinflammation with NSAIDs was one of the earliest approaches to treat various related disorders. However, despite its strong therapeutic rationale, previous clinical trials investigating compounds with anti-inflammatory properties, including NSAIDs, failed to achieve primary efficacy endpoints. In fact, there are bibliographic reports mentioning that NSAIDs are not the best pharmacological option [129], due to the absence of clear effects on the inflammatory response inhibition, which leads to neurodegeneration, as demonstrated by the lack of significant results of these drugs on cognitive impairment and the severity of AD in long term.

Current treatments can address individual symptoms for some disorders, but there is no known cure for any neuroinflammatory disorder. While the cause, pathology and symptoms of these disorders are extremely diverse, they all share a core inflammatory component [130].

Recent drug development projects were based on the emergence of new potential targets in different genomic and proteomic studies. Despite all of the drug development efforts undertaken, the number of successful drugs and novel targets have been lower than expected during the past few decades [130,131]. Therefore, it is necessary to develop other anti-inflammatory drugs, such as those derived from medicinal plants, efficient in combating neuroinflammatory processes.

## 7. Conclusions

The main goal of neuroinflammatory disease therapies is to stop or slow down the progression of the pathology. Current treatments include antioxidants and NSAIDs, among other substances; however, they have a modest clinical effect in the treatment of symptoms and do not prevent the progression of dementia, even though they may improve some cognitive functions.

Currently, there are data that species of the *Agave* genus have pharmacological properties as immunomodulators in different biological systems; however, only one study has demonstrated their ability to reduce neuroinflammation, which is complicated by the presence of the blood–brain barrier.

More pharmacological and chemical studies are needed to evaluate the possible therapeutic effects of the compounds present in the leaves of *Agave* species, specifically on neurodegenerative conditions, in which they exert beneficial functions in biological phenomena linked to inflammation and oxidative stress, among other factors.

*Agave* species, besides being an important source of pharmacologically active products and a cultural staple food in Mexico, have the potential to become a sustainable resource for the production of phytopharmaceuticals. Through research, mainly using the leaves, which are usually discarded and considered as by-products, despite possessing a wide variety of components such as saponins, their immense potential as a raw material for the development of new therapeutic resources can finally be implemented to elucidate possible applications in neuroinflammatory conditions.

## Figures and Tables

**Table 1 plants-11-02208-t001:** Traditional uses, common names and distribution of the *Agave* species mentioned in the text.

Scientific Name	Common Name (s)	Uses	Distribution in Mexico	References
*A. offoyana* Jacobi., 1864	No data found	No data found	No data found	[1]
*A. angustifolia* Haw	“Maguey espadín” “Maguey de monte” “Bacanora” Caribbean *Agave*	Urticaria Dysentery pain of wounds relieve rheumatic pain inflammation. Mezcal production	Oaxaca Durango	[9]
*A. rzedowskiana* P. Carrillo, Vega & R. Delgad.	No data found	No data found	Nayarit/Sinaloa/Jalisco	[21]
*A. ornithobroma* Gentry, 1982	“Maguey Pajarito”, “Amole”, “Lechuguilla”.	Fiber for the manufacture of backpacks and mats	Nayarit/Sinaloa	[21]
*A. salmiana* Otto ex Salm-Dyck	“Maguey pinto”, En Michoacán: “Akamha” (purépecha *) “Maguey cimarrón” “Maguey pulquero” “Pulque *Agave*”	Inflammation, gastritis, diabetes, wounds, blows and cough	Tlaxcala, Hidalgo, Querétaro, San Luis Potosí y Zacatecas.	[22,23]
*A. barbadensis* Trel	No data found	No data found	No data found	[21]
*A. impressa* Gentry	“Maguey masparillo” “Lechugilla”	Ornamental	Sinaloa Durango Nayarit	[22]
*A. marmorata* Roezl	Maguey de caballo Pitzomel	Cough Asthma internal bumps Ornamental.	Puebla Oaxaca	[22]
*A. intermixta* Trel	No data found	It is used as a decoction of the plant to treat cancer problems	No data found	[22]
*A. schidigera* Lem.	“Lechugilla mansa” Thread-leaf *Agave*	To relieve gastric problems in children	Sonora y Chihuahua a Michoacán, Jalisco, Aguascalientes, San Luis Potosí	[22,24]
*A. americana* L., 1753	“Maguey pinto”, “Mezcal”, American *Agave*	Gastric ulcers, eye inflammation, diuretic, wounds, diarrhea. Ornamental	Tropical America, widely distributed in Mexican territory	[24]
*A. tequilana* F.A.C. Weber	“Maguey azul” Blue *Agave*	Tequila production	Jalisco, Michoacán, Guanajuato y Nayarit. Tamaulipas	[25]
*A. fourcroydes* LEM.	“Henequen”	It is a natural fiber used to make hammocks and mats. It is fermented to produce alcohol (similar to mezcal)	Yucatán	[26]
*A. utahensis* Engelm	Utah *Agave*	Nursery Stock Product	No data found	[27]
*A. schotti* Engelm	“Maguey” Arizona shin-dagger “Amol” (Náhuatl) **	No data found	Chihuahua, Sonora, and Baja California.	[28]

* Purépecha: a national indigenous language of the Tarascan linguistic family, which is spoken in Michoacan, Mexico; ** Náhuatl: an indigenous language of the Uto-Aztecan linguistic family, which is spoken in Mexico and Central America.

**Table 2 plants-11-02208-t002:** Chemical structures of compounds with anti-inflammatory activity contained in *Agave* species.

Natural Compound Name	Number	Chemical Structure
smilagenin-3-*O*-[β-D-glucopyranosyl (1→2)-β-D-galactopyranoside]	1	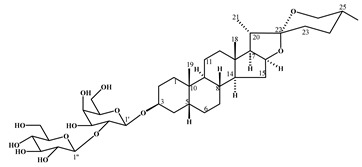
3-*O*-[(6’-*O*-palmitoyl)-β-D-glucopyranosyl sitosterol	2	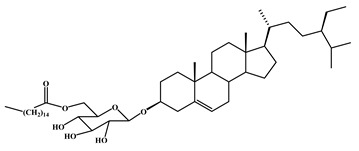
Agavesaponin E	3	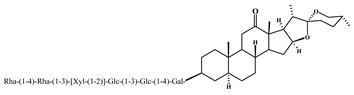
Agavasaponin H	4	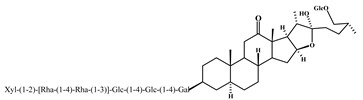
Hecogenin	5	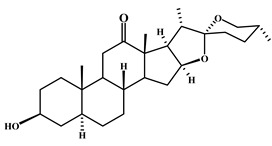
Tigogenin	6	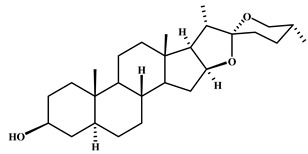
β-sitosterol	7	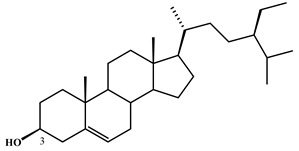
Isorhamnetin	8	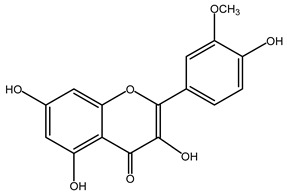
Hesperidin	9	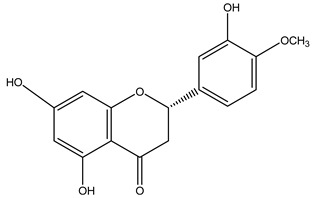
Delphinidin	10	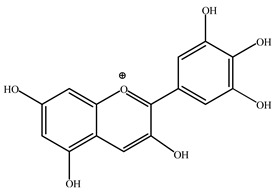
Quercetin	11	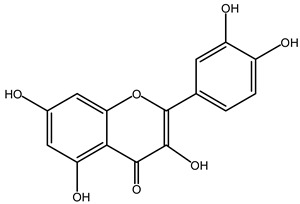
Apigenin	12	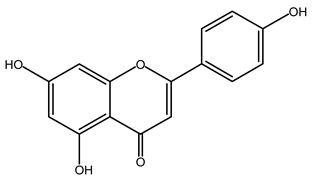
Catechin	13	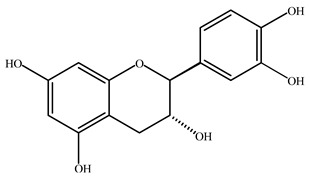
Ginsenoside Rg1	14	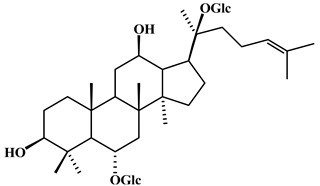
Ginsenoside Re	15	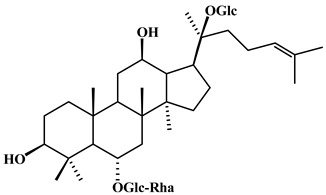

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
