# Peer review of "Potential Use of Agave Genus in Neuroinflammation Management"

_plants, 2022, doi:10.3390/plants11172208_

Round 1

Reviewer 1 Report

Review papers always get attention, but a survey of previous work should generate new insights which are missing in present work. The submitted manuscript can be considered as the initial review material on the basis of which the Authors could develop an overview of the certain characters of Agave species and their use.

The manuscript contains many inaccuracies that make it not acceptable for the publication.

The title of manuscript is illogical. Phytochemistry and pharmacology are separated as independent subjects affecting   neurinflammation. 

The manuscript does not have a clear purpose and objectives.

The content of manuscript is inconsistent, does not match titles of sections.

In the section 2 mixed different things as chemistry, biological activity and products that should be separated and considered for the uses and potential application of Agave spp.  

In the section 3 presented only Table 2 without any interpretations and discussion.  The following sections described some metabolites with their biological activities; however, this does not correspond to the content given in the manuscript.  Chemical composition and  activities  were declared in the sections 2 -3.   

The titles of the Tables 1 and 2 must be changed and the tables themselves have been rearranged to be sufficiently logically informative.

If the Authors aimed to review Agave spp. importance in the treatment of neuroinflammatory diseases, the entire content of the manuscript with the presented data and their appropriate discussion should be aimed at revealing this topic.

It is desirable to reveal the mechanism of action of  Agave spp. through the accumulation of specialized metabolites and to identify species that have the greatest potential for target biological activity.

The genus name must be written in italics. The manuscript needs to be English edited.

Author Response

Dear reviewer,

Dear reviewer, we are attaching a pdf document with the answers to your kind comments.

Reviewer 2 Report

The effort made by the authors is very valuable, as nowadays it is impossible to follow most of the current literature on a topic of interest. This paper is a comprehensive review focused on the biologically active compounds of Agave species and their effects on neuroinflammation and associated diseases. The manuscript fits within the scope of the journal. The manuscript is interesting and the idea is nice. The author’s work on discussing achieved results is appreciated. The revisions are necessary to improve the clarity of the presentation.

I have some recommendations for authors:

Please rewrite the title so that it is grammatically correct and easy to understand.

Please include some information about the work method: How do you search literature data? How was a period? Which sources?

What is the novelty and originality of the review, considering that recent articles on this topic are published?

Please remove the abbreviations from the text for the scientific name of the plants. It makes reading very difficult.

The text contains many concrete results, without any discussion of the authors based on these results. A review is not just a series of results obtained by other authors. It must show the personal opinion of the authors of the review.

Check the bibliography to be written according to the requirements of the journal.

Author Response

Dear reviewer, we are attaching a pdf document with the answers to your kind comments.

Reviewer 3 Report

The natural products from the agro-industrial waste of Agave plants have a high pharmacological potential. This review addresses medicinal uses, pharmacology and chemical studies of Agave species, but this review is neither exhaustive nor novel. There is a review for this recently, see ref 90 of the paper.

On the other hand, the authors describe the neuroantiinflamatory activity literature for the families of compounds present in Agave. Sitosterol derivatives, are substances that are present in all plant species, so they are not characteristic of this genus. Steroidal saponins are the metabolites most frequent described in Agave (spirostanics and furostanics). Ginsenosides and triterpene saponins are quite distant structurally. The furostanic saponins were the only saponins that can be close to those of Agave were isolated from Smilax davidiana or S. china. Finally, only one compound from this genus has been tested for this activity, cantalasaponin-1.

Paper should be included:

Antiinflamatory and modulatory effects of steroidal saponins and sapogenins on cytokines: A review of pre-clinical research. Phytomedicine 96, 153842 (2022).

This review is directed at neuroinflammatory activity, however, there is too little literature on neuroinflammatory activity in natural products of the Agave genus to be reviewed.

In general, there are mistakes in style of natural product cientific names and such as indicating as saponins, compounds that are aglycones, or including a herbicidal activity as a pharmacological activity.

Author Response

(The authors gave the same response as above.)

Reviewer 4 Report

Subject of this publication is interesting, however the article requires rethinking in terms of structure and selection of content. Moreover, there is a lack of coherence and thematic relationships.

1.    At the end of subsection “Introduction” the Authors should describe aim of review article.
2.    This article do not contains description of methods. Methods used in review should include parameters such as exclusion and inclusion criteria, kind of articles (review or experimental studies) and key words. Description should include number articles selected to this review, which have been analyzed in sections of article. Moreover, the Authors should inform about progression in number of publications.
3.    Table 1 contains mistakes in scientific name of plants.
4.    At line 224. The title “Natural Agave products” should be changed.
5.    At 320 and 341- subsections are not an integral part of the work.
6.    The conclusions are not in the main topic of the article.

Author Response

(The authors gave the same response as above.)

Round 2

Reviewer 1 Report

The content of the newly submitted manuscript is positively modified, but the authors must revise the manuscript again to ensure that the content and form have a logical structure.

The sources not provided that support the consideration “Since the process of neuroinflammation can be a start point and of several chronic neurodegenerative diseases... “

Please correct the statement “secondary metabolites isolated from the genus Agave..“ because metabolites cannot be isolated from a genus but from plants of the corresponding species of the genus.

Please note throughout the manuscript that the review deals with the potential use of the species of Agave genus.   

Please modify illogical statement “Therefore, studies showing their antioxidant, anti-inflammatory, cytotoxic and antibacterial effects, among others, are mentioned”.

The review covers various aspects of research on agave species, but there is much logical confusion in the order in which they are arranged and presented.

Please change the title of the section “3. Generalities of the genus Agave” in which the biological activity of to a more appropriate one as.

Please change the title of the chapter "3. General aspects of the genus Agave", which included a review of biological activities of the species, to a more appropriate one.

The logical order of the following sections and their titles must be highly revised.

The name of genus Agave must be written in italics throughout the manuscript including the title.

 In the conclusions was used “magueys” the meaning of which was not described above in the text of manuscript.

Author Response

Title: “Potential use of Agave genus in neuroinflammation management”

Reference number: plants-1790008

The authors thank the referees for the time invested in reviewing the work. All changes, are marked in green within all the text.

Reviewer 2 Report

All comments have been raised accordingly

Author Response

(The authors gave the same response as above.)

Reviewer 3 Report

In this second version of the review "Potential use of the Agave genus in the management of neuroinflammation" a better definition of the objective of the review is offered.

There is only one reference on neuroinflammatory activity in Agave species, so does not make sense a revision about it. In this way, it is proposed that the review synthesize the existing literature on the biologically active compounds of Agave species that are related in some way to inflammation.

In the search planning it is indicated that the keywords were “anti-inflammatory Agave” plus polysaccharides (2 publications), saponins (195 publications), triterpenes (91 publications), alkaloids (200 publications), flavonoids (2 publications) and fatty acids (258 publications), give rise to a good number of publications. Only two sections with biologically active compounds are included, paragraph 4. “Sitosterol” (is not included sitosterol or phytosterols as a keyword) and paragraph 5. Saponins, the other biologically active compounds are not described.

On the other hand, the vast majority of the manuscript is made up of generalities of the Agave genus and biological bases of neuroinflammation, which, although it may be interesting as an introduction to justify the subject of the review, do not come from the results of search planning, and it is information that does not bring novelty.

Biological activities that do not have to be related to anti-inflammation are described, and some activities that have been extensively described, such as antifungal activity, do not appear (Yang, Antimicrob. Agents. Ch., 2006, 50, 1710-1714).

Therefore, although a review of the anti-inflammatory potential of species of the Agave genus may be interesting, the results offered are not related to the objective or the methodology followed for the review.

Author Response

(The authors gave the same response as above.)

Reviewer 4 Report

The authors correctly respond to comments from the reviewer.

The manuscript has been revised. 

Author Response

(The authors gave the same response as above.)

Round 3

Reviewer 3 Report

This third version presents more information of anti-inflammatory substances that are present in the extracts of Agave species. The authors indicate in coverletter that: “… based on the reviewers proposal, in order to complement and support the review, we added information on Agaves chemical composition, emphasizing some data on the antineuroinflammatory activity of compounds isolated in other species, but also known to be found in Agaves”

 In this case, the authors have included some flavonoids that have shown anti-inflammatory effects.

 The bibliographic search that the authors have carried out includes a good number of references to triterpenes (91 publications), alkaloids (200 publications) and fatty acids (258 publications), but information on compounds of these groups is not included in the review. An explanation would be necessary to know why this information is not included in the review, in the event that specific molecules were not described in the references, but rather come from an analysis of the types of metabolites present in the extracts.

 On the other hand, a section called “terpenoids” is included, which is not consistent with the search of the review that has been carried out in any case of triterpenoids. Coherence between the method and the results would be necessary.

 Since other biological activities have been maintained, in the case of herbicide activity there is a recent publication, Agronomy 2021, 11(12), 2404; https://doi.org/10.3390/agronomy11122404.

IUPAC natural product names should have the O of the glycosidic bond in italics and the D or L of the sugar configuration in lowercase and small caps.

Sometimes saponins and sapogenins are described under the term saponins, the two groups of compounds should be differentiated in the text. On other occasions it is indicated that certain sapogenins are found in extracts of Agave species, which is not usually, rather these sapogenins come from the hydrolysis of the saponins present, this fact should be reviewed.

For the saponins of A. fourcroides, compound 1 cannot be indicated, that is the name of the saponin in the original reference, it would be necessary to put its scientific name.

On another occasion it is indicated that tigogenin is a saponin, and it is a sapogenin.

For A. uthaensis a saponin is named as smilagenin diglycoside, please indicate the correct scientific name of the saponin.

For greater ease of reading, it would be interesting to include a figure with the structures of the secondary metabolites described in the text, at least the most representative, numbered and to include that numbering in the text.

Author Response

Thank you very much, your comments contributed to improve the manuscript. 

Responses to your comments are attached in the pdf file below.
